# Effects of Vitamin D Receptor, Metallothionein 1A, and 2A Gene Polymorphisms on Toxicity of the Peripheral Nervous System in Chronically Lead-Exposed Workers

**DOI:** 10.3390/ijerph17082909

**Published:** 2020-04-23

**Authors:** Hsin-Liang Liu, Hung-Yi Chuang, Chien-Ning Hsu, Su-Shin Lee, Chen-Cheng Yang, Kuan-Ting Liu

**Affiliations:** 1Division of Internal Medicine, Department of Emergency Medicine, Kaohsiung Medical University Hospital, Kaohsiung Medical University, Kaohsiung 80708, Taiwan; cutelpliu@gmail.com; 2Department of Public Health, Kaohsiung Medical University, Kaohsiung 80708, Taiwan; ericch@kmu.edu.tw; 3Department of Environmental and Occupational Medicine, Kaohsiung Medical University Hospital, Kaohsiung 80708, Taiwan; abcmacoto@gmail.com; 4Department of Pharmacy in Kaohsiung Chang Gung Memorial Hospital, and School of Pharmacy, Kaohsiung Medical University, Kaohsiung 80708, Taiwan; chien_ning_hsu@hotmail.com; 5Center for Stem Cell Research, Kaohsiung Medical University, Kaohsiung 80708, Taiwan; k831702000@gmail.com; 6School of Medicine, College of Medicine, Kaohsiung Medical University, Kaohsiung 80708, Taiwan

**Keywords:** vitamin D receptor, metallothionein 1A and 2A, gene polymorphisms, lead toxicity, sensory nervous system

## Abstract

Chronic exposure to lead is neurotoxic to the human peripheral sensory system. Variant *vitamin D receptor (VDR)* genes and polymorphisms of metallothioneins (MTs) are associated with different outcomes following lead toxicity. However, no evidence of a relationship between lead neurotoxicity and polymorphisms has previously been presented. In this study, we investigated the relationship between the polymorphisms of *VDR*, *MT1A*, and *MT2A* genes and lead toxicity following chronic occupational lead exposure. We measured vibration perception thresholds (VPT) and current perception thresholds (CPT) in 181 workers annually for five years. The outcome variables were correlated to the subject’s index of long-term lead exposure. Polymorphisms of *VDR*, *MT1A*, and *MT2A* were defined. The potential confounders, including age, sex, height, smoking, alcohol consumption, and working life span, were also collected and analyzed using linear regression. The regression coefficients of some gene polymorphisms were at least 20 times larger than regression coefficients of time-weighted index of cumulative blood lead (TWICL) measures. All regression coefficients of TWICL increased slightly. *MT1A* rs11640851 (AA/CC) was associated with a statistically significant difference in all neurological outcomes except hand and foot VPT. *MT1A* rs8052394 was associated with statistically significant differences in hand and foot CPT 2000 Hz. In *MT2A* rs10636, those with the C allele showed a greater effect on hand CPT than those with the G allele. Among the *VDR* gene polymorphisms, the Apa rs7975232 (CC/AA) single nucleotide polymorphism was associated with the greatest difference in hand CPT. *MT2A* rs28366003 appeared to have a neural protective effect, whereas Apa (rs7975232) of *VDR* and *MT2A* rs10636 increased the neurotoxicity as measured by CPT in the hands. *MT1A* rs8052394 had a protective effect on large myelinated nerves. *MT1A* rs11640851 was associated with susceptibility to neurotoxicity.

## 1. Introduction

Lead is widely used in industry and remains a public health concern despite leaded gasoline being phased out in Taiwan in 2000 [1,2]. Recent epidemiological and toxicological studies have reported that lead exposure is associated with several diseases affecting the cardiovascular system [3] and renal system [4,5], as well as the hepatic system [6,7]. In addition, lead is capable of inducing oxidative stress that affects pregnancy and the life course [8,9,10]. Mani et al.’s study assessed the ecogenetics of lead toxicity and its influence on risk assessment [11]. The blood lead levels in lead-exposed workers have been reported to be between 20 and 50 μg/dL in Taiwan [12]. The peripheral nervous system is one of the target systems in lead intoxication. Workers with 5 year mean blood lead levels greater than 30 μg/dL had higher vibration perception thresholds (VPTs) [13] than those who had lower blood lead levels. Moreover, using sensory nerve current perception thresholds (CPTs), we revealed that drinking milk (approximately 700 mL per day) might protect against lead peripheral neurotoxicity [14]. Lead–calcium interactions are probably the most studied nutritional factors affecting lead toxicity, both clinically [15,16] and experimentally [17,18] 

The toxicological relationship in which lead affects calcium and vitamin D metabolism is not well understood. The biological interactions between lead and calcium are complex [19]. Some authors suggest that variant vitamin D receptor (*VDR*) alleles (e.g., Bsm rs1544410, Apa rs7975232 and Taq rs731236) modify lead concentrations in the blood and bone either by affecting lead and/or calcium content [20,21,22,23,24,25,26]. Chuang et al. found that chronically lead-exposed workers had higher blood lead levels and index of cumulative blood lead (ICL) in the Apa AA genotypes. [27]

Metallothioneins (MTs) are cysteine-rich, metal-binding, low-molecular-weight proteins [28,29]. MT proteins have multiple functions, such as free radical scavenging and the detoxification of metal(loid)s including cadmium, arsenic, mercury, and lead [30,31,32,33,34]. The human *MT* genes consist of four subfamilies, classified as *MT1*–*MT4*. *MT1* and *MT2* are expressed in all organs, whereas *MT3* is mainly expressed in brain tissue and *MT4* is expressed in differentiating stratified squamous epithelial cells [30,32]. *MT1* and *MT2* are isoforms, consisting of sub-isoforms coded by various functional genes. In addition, *MT1A* and *MT2A* have anti-oxidative effects [35], but the effect varies between individuals despite similar exposure levels. One of the reasons for this is genetic polymorphisms. Two genetic polymorphisms of *MT1A* (rs11640851 and rs8052394) and two of *MT2A* (rs10636 and rs28366003) are defined [36].

To the best of our knowledge, this is the first study on the relationship between the polymorphisms of *VDR*, *MT1A*, and *MT2A* and lead effects on the sensory nervous system in humans. The aim of this study was to investigate the association of blood lead level, VPT, and CPT in chronic occupational lead-exposed workers, and to assess whether the association was influenced by *VDR*, *MT1A*, and *MT2A* gene polymorphisms.

## 2. Materials and Methods

### 2.1. Study Population and Study Design

The study was from a lead–acid battery factory, where workers have been followed up since 1990 with annual health examinations. The total number of workers was 217 (36 workers were excluded due to organic solvent exposure, head or nerve injuries, major neurological diseases, diabetes, or personal withdrawal), and we studied their peripheral nervous systems [13]. 

The study was approved by the institutional review board in Kaohsiung Medical University Hospital to confirm that it was performed in accordance with relevant guidelines and regulations, and informed consent was obtained from all participants. The approval number from institutional review board in Kaohsiung Medical University Hospital was KMUHIRB-20120010. Finally, 181 workers consented to participate in our study.

### 2.2. Blood Lead Measurement and Accumulation Indicators

Blood lead levels were analyzed by Zeeman effect graphite furnace atomic absorption spectrometry (GF-AAS, Perkin-Elmer 5100 PC with AS 60 autosampler). All coefficients of variation (CVs) were <3% for blood lead measurements at high (70.5–82.7 μg/dL) and medium (37.1–45.3 μg/dL) levels, and were <5% for those at low levels (5.6–8.9 μg/dL), using commercial standards as intra-laboratory quality controls (Betherning Institute, Bio-Rad). Our laboratory participates in an inter-laboratory blood-lead proficiency-testing program of the Centers for Disease Control and Prevention and all our measurements were within range, indicating that our blood lead measurements were accurate.

The index of cumulative blood lead (ICL) was calculated by integrating blood lead concentrations over each worker’s length of occupational exposure using the trapezoidal rule [37].
ICL = ∫PbB_t_Δt = Σ0.5(PbB_i_ + PbB_i+1_)Δt_i_, expressed in year *μg/dL.
where PbB_i_ and PbB_i+1_ represent the ith and (i +1)th measurements of blood lead, taken Δt years apart.

Time-weighted index of cumulative blood lead (TWICL) was calculated by dividing ICL by the summation of employment time:$$\mathrm{TWICL}=\frac{\Sigma 0.5\left(PbB_{i}+PbB_{i+1}\right)\Delta t_{i}}{\Sigma \Delta t_{i}}$$

Reliable measurements of blood lead began in 1991. Of the workers, 45% had entered the factory before 1990, when we began to follow up their health status. Workers with no existing records on blood lead concentrations before 1990 were assigned a value of 15 μg/dL from birth to the date of employment. This value was based on the mean concentration of blood lead in a population survey of Taiwan in 1984 (Department of Health, 1984). For each worker who had been hired before 1991, the first value of the blood lead level measured by our laboratory (usually in 1991) was assumed to be representative of the blood lead levels from 3 months after the date of employment until the date of the first blood lead measurement. We assumed that blood lead levels had gradually increased to this value from the date of employment. In fact, the industrial hygiene in the plant had changed little until we intervened in 1990; thus, before 1991, blood lead levels in workers at this plant could be rationally assumed to have reached equilibrium, similar to the levels at first measurement in our laboratory.

### 2.3. Vibration Perception Threshold Test

VPT was measured with a vibrameter (Technoque, Tokyo, Japan), with the frequency of vibration fixed at 220 Hz and the unit of measurement one-hundredth gravity (10^−2^ g, or 0.098 m/s^2^). The intensity of stimulation ranged from 0 (minimum) to 999 (maximum). Two sites, the distal phalanx of the left index finger and the left big toe, were tested for each worker. A standard protocol was used, which was called “method of limits yes–no procedure” and showed an acceptable measure of reproducibility and validity [38]. During the test, a practice trial test was administered to ensure that each worker understood the test. Stimuli were then gradually increased from the minimum until each worker reported feeling a vibration sensation, and the score of intensity unit was recorded. The mean of three tests for a site represented the VPT value of the site. Room temperatures were maintained above 27 °C. All workers were tested by the same technician trained in neurological examination. This method was previously successfully applied in a study of workers with exposure to styrene in Taiwan [39].

### 2.4. Current Perception Threshold Test

Quantitative sensory tests, CPTs, were carried out using an electric neurophysiological device, Neurometer (Neurotron Co., Baltimore, MD, USA). They are non-invasive, painless, and require approximately 15–20 min per test. To avoid the probability of being affected by handy vibration tools, two sites, the distal phalanx of the left index finger and the left great toe, were tested for each participant. More than 99% people in Taiwan are dominant users of the right hand. All the CPT measurements were completed by a trained technician, who measured CPTs from 0.01 (mA) initially, increasing until the worker reported a sensation of tingling, buzzing, or warmth. The current was then immediately turned off. Trials at various amplitudes were used to determine the minimum threshold, following which the score of current was recorded in a unit of 0.01 mA [40]. Three tests were performed on each site at each frequency and the means were recorded as the final CPT at that site at each frequency for each worker. A neurometer distinguished different types of nerve fiber damage by using different frequencies (5, 250, and 2000 Hz) of sinusoidal electric stimulus. The frequency of 5 Hz stimulates the C fibers, and 250 and 2000 Hz were used for A-delta and A-beta, respectively (AAEM Equipment and Computer Committee, 1999). In the tests, higher threshold values indicated less sensitivity of sensory nervous fibers. During the test, room temperatures were over 27 °C.

### 2.5. Questionnaire

A short questionnaire on job title, medical and working history, educational status, and alcohol and cigarette consumption was given before the health examination. The workers also reported their frequency of smoking and drinking per week during the previous three months; the questionnaire was reviewed and checked for completeness by an occupational physician when the annual health examination was performed. Although none of the workers ever used a portable vibration tool (such as a hand-carry electric saw) at work, some occasionally cut lead plates with static electric saws fixed on the working tables. In the questionnaire, each worker was asked whether such vibration tools were used at work. The use of vibration tools was coded as a binomial variable (yes versus no) to model its effect in our analysis. None of the workers ever used vibration tools involving their feet.

### 2.6. DNA Isolation and Storage

Genomic DNA was isolated from peripheral blood using QIAamp DNA Blood Mini Kits (Qiagen, Valencia, CA, USA), and the final preparations were stored at −20 °C for use as templates in PCR.

### 2.7. VDR Genotyping

The genotypes for three restriction-fragment-length polymorphisms of the *vitamin D-receptor* (*VDR*) gene were determined by PCR amplification and enzymatic digestion of the products with Apa (rs7975232), Bsm (rs1544410), and Taq (rs731236). The primers and restriction endonucleases used to identify the polymorphism are commercially available, for example, Apa, Taq (Boehringer Mannheim Corp., Indianapolis, IN), and Bsm (New England Biolabs Inc., Beverly MA). The forward primer for the Apa and Taq polymorphisms was the same as that used for amplification of the Bsm polymorphism: 5′-CAACCAAGACTACAAGTACCGCGTCAGTGA-3′. The reverse primer used for the Apa and Taq polymorphisms is located in Exon 9: 5′-GCAACTCCTCATGGGCTGAGGTCTCA-3′ and 5′- AACCAGCGGGAAGAGGTCAAGGG-3′ for the Bsm. PCR was performed with a Biometra Trio thermoblock (Floral City, Fla.) under standard conditions for 35 cycles, and with 65 °C as the annealing temperature. The PCR product for the Bsm polymorphism was 825 base pairs (bp) long, and the restriction fragments were 650 bp and 175 bp long. The PCR products for the Apa and Taq polymorphisms were 2000 bp long; the lengths of the fragments after digestion with Apa were 1700 and 300 bp, and the lengths of the fragments after digestion with *Taq* were 1800 and 200 bp.

### 2.8. MT1A and MT2A Genotyping

The dbSNP database (http://www.ncbi.nlm.nih.gov/SNP/) of the National Center for Biotechnology Information and the HapMap database were used to search for single nucleotide polymorphisms (SNPs). Four SNPs (*MT1A*: rs11640851 and rs8052394; *MT2A*: rs10636 and rs28366003) were analyzed. We used TaqMan allelic discrimination assays (Applied Biosystems, Foster City, CA, USA) to genotype the SNPs, and the results were read with an Applied Biosystem 7300 Real-Time PCR System (Life Technologies Corp., Carlsbad, CA, USA). In brief, the genotyping of the *MT1A* and *MT2A* polymorphisms (rs11640851, rs8052394, rs10636, and rs28366003) was performed via TaqMan SNP genotyping assays (Applied Biosystems, Foster City, CA, USA) [41]. The final volume for each reaction was 10 μL, containing 5 μL of TaqMan Universal PCR Master Mix, 0.25 μL of primers/TaqMan probe mix, and 10 ng of genomic DNA. The real-time PCR comprised an initial denaturation step at 95 °C for 10 min, followed by 40 cycles each at 92 °C for 15 s and 60 °C for 1 min. Fluorescence was measured with an Applied Biosystems StepOne Real-Time PCR System (Life Technologies Corp., Carlsbad, CA, USA). The allele frequencies were determined using ABI SDS software. Genotyping was repeated on a random 10% sample to confirm the results of the original analysis.

### 2.9. Statistical Analysis

Descriptive statistics were used to calculate the arithmetic means of continuous variables, including the blood lead concentrations, age, height, weight, working history, and working life span. Descriptive statistics were also used to present the dispersion of these data. For category variables, such as sex, smoking, and alcohol consumption, proportions were used. The characteristics, neurological outcomes, and genotypes of these workers were divided into four groups according to quartiles of TWICL levels. The chi-squared test was used to present the difference of nominal variables among groups. Furthermore, if the expected numbers were below 5, Fisher’s exact test was used. ANOVA was used to test for the difference of continuous variables (age, weight, height, working history) among TWICL quartile groups. If the ANOVA tests were significant, then post hoc test (Scheffe) was used to compare the different. Depending on their genotypes, the participants were divided into three groups. The distribution of CPT and VPT were bell-shaped; thus, we used raw data for regression analysis. Multiple linear regression analysis was used to evaluate the association of TWICL and the neurological outcomes while adjusting for genotype and other variables such as age, sex, smoking status, height, work life span, and drinking status. Hand VPT and CPT were adjusted for the use of hand vibration tools. IBM-SPSS 19 statistical software was used for data analysis, the tests were two-tailed, and thus, a *p*-value less than 0.05 was considered significant.

## 3. Results

The total number of qualified workers examined by CPT testing was 181. The difference in the mean blood lead levels between participants (n = 181, mean ± S.D. = 25.5 ± 12.7) and non-participants (n = 36, mean ± S.D. = 28.0 ± 13.8) was not significantly different; thus, we suspect that the exclusion of these study participants did not bias our results. The characteristics of these workers are shown in Table 1. The distribution of sex, alcohol consumption, smoking, and vibration tool use were significantly different among different quartiles of TWICL by chi-square and Fisher’s exact tests. More male workers were noted in the 75–100 percentile. Similarly, the body height (cm) and working history (year) were significantly the highest in the 75–100 percentile group by the ANOVA and post hoc (Scheffe) test.

Table 2 shows the neurological outcomes of these workers. Statistically significant differences among the quartile groups were observed for all tested neurological outcomes, except for hand VPT. Significantly higher thresholds for foot VPT and both foot and hand CPT at all frequencies were observed for individuals in the highest TWICL quartile (75–100 percentile) by ANOVA and post hoc (Scheffe) tests. This result is consistent with previous studies [13,14].

The genotype distributions of three *VDR*, two *MT1A*, and two *MT2A* SNPs, within each TWICL quartile, are shown in Table 3. The minor allele frequencies for these SNPs were 12% for *VDR*-Bsm, 32% for *VDR*-Apa, 8% for *VDR*-Taq, 46% for *MT1A* rs11640851, 38% for *MT1A* rs8052394, 26% for *MT2A* rs10636, and 8% for *MT2A* rs28366003. No SNPs significantly violated Hardy–Weinberg equilibrium. For *VDR* Bsm (rs1544410), *MT1A* (rs11640851), and *MT2A* (rs10636), higher percentages of individuals with variant alleles were observed in higher quartiles of TWICL levels.

We used multiple linear regression analyses to investigate the effects of different genotypes on neurological outcomes with TWICL and controlled for other confounders, as shown in Table 4. TWICL without gene adjustment was significantly associated with all tested neurological outcomes except hand CPT 250 Hz and 2000 Hz. Among the *VDR* SNPs, Apa rs7975232 (CC/AA) had the largest influence on all three frequencies of hand CPT. *MT1A* rs11640851 (AA/CC) was associated with a significant difference in all neurological outcomes except hand and foot VPT. *MT1A* rs8052394 (GG/AA) was associated with a significant difference in hand and foot CPT at 2000 Hz. In *MT2A* rs10636, the C allele had a greater influence on all three frequencies of hand CPT when comparing CC with GC and GG. *MT2A* rs28366003 (AG/AA) was associated with a significant difference in all neurological outcomes except foot VPT and had negative regression coefficients. Two additional neurological outcomes (hand CPT 250 Hz and 2000 Hz) were associated with a significant difference after adjusting for *MT2A* rs28366003 (AG/AA), and all regression coefficients of TWICL increased at the same time. The regression coefficients of some gene polymorphisms were at least 20 times larger than the regression coefficients of TWICL. The interactions of SNPs and TWICL were tested but not significant; the actual effect of SNPs on levels of TWICL was not tested here.

## 4. Discussion

To the best of our knowledge, this is the first study on the relationship among the polymorphisms of the *VDR*, *MT1A*, and *MT2A* genes and lead toxicity on the peripheral nervous system in humans. One hundred and eighty-one occupationally exposed workers participated in this study between 1990 and 1995.

In regression models, the regression coefficients of some gene polymorphisms were at least 20 times higher than regression coefficients of TWICL. TWICL needed to increase by at least 20 μg/dL to have the same influence on neurotoxicity compared with other gene polymorphisms.

Very few studies have discussed the influence of *VDR*-Apa (rs7975232) SNP. A previous study mentioned that higher TWICL was noted for the CC genotype than CA type than AA type in Apa (rs7975232) [27]. A similar trend was also observed in this study, although the difference was not significant. Moreover, all three frequencies of hand CPT were significantly higher for the CC versus the AA genotype of Apa. Chuang et al. showed an independent neurological protective effect of milk intake on hands, but not on feet, while adjusting blood lead levels, suggesting that the protective mechanism is not solely based on decreasing intestinal absorption [14]. Further studies are required to investigate the relationship between Apa (rs7975232) CC genotype and calcium–lead interactions. 

Yang et al. found that the G allele variants of *MT1A* rs8052394 caused a decrease in superoxidase dismutase (SOD) activity in diabetes patients, although this was not significant in controls [35]; thus, it could deplete SOD. We found that *MT1A* rs8052394 GG type had protective effects in hand and foot CPT 2000 Hz compared to AA type. CPT 2000 Hz stimulates the A-beta large myelinated fibers. One study found that there is atrophy of large myelinated axons in Metallothionein-I, II-knockout mice [42]. The reduction in axon caliber is likely to be due to the effects of reactive oxygen species (ROS) on cytoskeletal components of axons such as neurofilaments. *MT1A* rs8052394 GG type would have a protective effect on large myelinated nerve due to its anti-oxidative effect.

The G allele of *MT2A* rs28366003 increased sensitivity to lead neurotoxicity, as measured by CPT, compared to the A allele. Pregnant women with the AG genotype (*MT2A*) might have high blood lead levels [43]. Individuals with the GG genotype had statistically lower Zn concentrations and higher Cd and Pb concentrations in the blood samples than individuals with AA and AG genotypes. The effect of Zn on Pb toxicity is controversial [44]. Further studies must be should be performed to identify the neuroprotective effects of Zn at high lead concentrations. The *MT2A* rs28366003 AA genotype may be associated with genetic susceptibility to neurotoxicity.

The C allele of *MT2A* rs10636 was associated with hyperlipidemia [35]. *MT2A* is intimately associated with oxidative stress [45]. A previous study showed that the increase in lipid peroxidation was found following lead oxidative stress, resulting in cell damage [46]. We found that the CC/GG genotype of *MT2A* rs10636 had a greater negative influence than the CG/GG genotypes on all three frequencies of hand CPT, but had no influence on the measures obtained from the feet. A possible reason is the longer nerve fibers are more susceptible to damage regardless of genotype. The peroneal nerves, with relatively longer fibers and a weaker blood barrier, are more susceptible to all physical risk factors and toxicants, including lead. Thus, the effect of lead on neurotoxicity as measured in the feet cannot be modified.

A limitation of this study was the relatively small sample size. This reduced the statistical power. Nonetheless, we still obtained significant results. The other limitation was that no multiple gene polymorphism analysis was performed, due to the relatively small sample size and the large number of possible gene combinations. 

## 5. Conclusions

In this study, we investigated the relationship between selected polymorphisms of *VDR*, *MT1A*, and *MT2A* genes and the impact of lead toxicity on the peripheral nervous system in chronically lead-exposed workers. TWICL needed to increase to at least 20 μg/dL to have the same effect on neurotoxicity compared to other known gene polymorphisms. The *MT2A* rs28366003 AA genotype may be associated with genetic susceptibility to neurotoxicity. The CC genotype of *VDR* rs7975232 (Apa) and the CC genotype for *MT2A* rs10636 increased neurotoxicity as measured by CPT in hands. The *MT1A* rs8052394 GG genotype had a protective effect on large myelinated nerves. The *MT1A* rs11640851 AA genotype was associated with increased susceptibility to neurotoxicity. The effect of interaction between metals such as zinc and lead could have a significant impact on neurotoxicity and may require future investigation.

## Figures and Tables

**Table 1 ijerph-17-02909-t001:** Characteristics of the lead workers participating in the study.

	Total	TWICL				*p*-Value
		≤25 Percentile	25–50 Percentile	50–75 Percentile	75–100 Percentile	
Number	181	45	45	46	45	
Sex						<0.001
Female	107 (59.1%)	35 (77.8%)	36 (80.0%)	31 (67.4%)	5 (11.1%)	
Male	74 (40.9%)	10 (22.2%)	9 (20.0%)	15 (32.6%)	40 (88.9%)	
Education						0.206
Elementary	54 (30.7%)	11 (26.8%)	16 (35.6%)	14 (31.1%)	13 (28.9%)	
Junior high	60 (35.2%)	12 (29.3%)	16 (35.6%)	17 (37.8%)	15 (33.3%)	
Senior high	41 (23.3%)	8 (19.5%)	8 (17.8%)	9 (20.0%)	16 (35.6%)	
Collage	20 (11.4%)	9 (22.0%)	5 (11.1%)	5 (11.1%)	1 (2.2%)	
Graduate	1 (0.6%)	1 (2.4%)	0	0	0	
Milk drinking	158 (87.3%)	40 (88.9%)	43 (95.6%)	38 (82.6%)	37 (82.2%)	0.185
Alcohol drinking	16 (8.8%)	0	1 (2.2%)	5 (10.9%)	10 (22.2%)	0.001
Smoking	42 (23.2)	5 (11.1%)	5 (11.1%)	5 (10.9%)	27 (60.0%)	<0.001
Vibration tool use	7 (3.9%)	0	1 (2.2%)	0	6 (13.3%)	0.002
Age (year)	39.5 ± 7.4	39.4 ± 7.2	39.7 ± 7.4	39.3 ± 7.5	39.7 ± 7.9	0.989
Body height (cm)	158.9 ± 8.2	157.7 ± 8.2	157.0 ± 8.6	158.6 ± 8.0	162.2 ± 7.2	0.012
Body weight (kg)	58.5 ± 10.6	56.0 ± 10.7	56.8 ± 11.0	60.3 ± 11.7	60.8 ± 8.2	0.076
Working history (year)	9.9 ± 7.5	7.1 ± 7.8	9.6 ± 7.5	9.7 ± 6.6	13.2 ± 7.2	0.001
Working life span (%)	24.3 ± 17.1	17.1 ± 17.6	23.8 ± 17.4	24.3 ± 15.1	31.9 ± 15.2	0.001

The total numbers of education level do not add to 181 due to missing data. TWICL: time-weighted index of cumulative lead.

**Table 2 ijerph-17-02909-t002:** Neurological outcome examinations of the lead workers.

	Total	TWICL				*p*-Value *
		≤25 Percentile	25–50 Percentile	50–75 Percentile	75–100 Percentile	
Number	181	45	45	46	45	
VPT (10^−2^ g)						
Hand	10.0 ± 6.0	8.4 ± 3.0	11.6 ± 7.8	10.5 ± 5.6	9.3 ± 5.9	0.111
Foot	18.9 ± 12.3	17.2 ± 8.8	16.7 ± 8.3	17.3 ± 10.0	24.9 ± 18.9	0.016
Hand CPT (10^−2^ mA)						
5 Hz	46.0 ± 19.2	41.70 ± 15.90	40.55 ± 16.08	49.82 ± 19.71	51.70 ± 22.46	0.008
250 Hz	76.9 ± 27.4	70.48 ± 24.21	70.02 ± 25.42	83.08 ± 24.27	83.96 ± 32.37	0.013
2000 Hz	205.7 ± 60.0	188.45 ± 54.56	192.65 ± 58.67	213.17 ± 52.33	228.22 ± 66.91	0.004
Foot CPT (10^−2^ mA)						
5 Hz	58.1 ± 26.1	48.59 ± 16.41	56.46 ± 29.58	60.92 ± 29.08	66.52 ± 24.35	0.009
250 Hz	109.6 ± 43.6	96.20 ± 30.77	98.22 ± 49.82	120.41 ± 46.76	123.30 ± 38.38	0.002
2000 Hz	270.9 ± 97.1	227.83 ± 69.74	253.55 ± 97.14	293.77 ± 95.32	307.90 ± 103.97	<0.001

* ANOVA and post hoc Scheffe tests. VPT: vibration perception thresholds; CPT: current perception thresholds.

**Table 3 ijerph-17-02909-t003:** SNP types in the lead workers.

	Number	Hardy–Weinberg Equilibrium *p*-Value	TWICL				*p*-Value *
			≤25 Percentile	25–50 Percentile	50–75 Percentile	75–100 Percentile	
*VDR*-Bsm (rs1544410)		0.067					0.024
GG	142 (79.3%)		36 (83.7%)	32 (71.1%)	39 (84.8%)	35 (77.8%)	
GA	32 (17.9%)		3 (7.0%)	12 (26.7%)	7 (15.2%)	10 (22.2%)	
AA	5 (2.8%)		4 (9.3%)	1 (2.2%)	0	0	
*VDR*-Apa (rs7975232)		0.076					0.109
AA	87 (48.6%)		25 (56.8%)	20 (45.5%)	24 (52.2%)	18 (40.0%)	
AC	68 (38.0%)		15 (34.1%)	22 (50.0%)	14 (30.4%)	17 (37.8%)	
CC	24 (13.4%)		4 (9.1%)	2 (4.5%)	8 (17.4%)	10 (22.2%)	
*VDR*-Taq (rs731236)		0.348					0.089
TT	153 (85.5%)		34 (77.3%)	36 (80.0%)	42 (93.3%)	41 (91.1%)	
TC	24 (13.4%)		8 (18.2%)	9 (20.0%)	3 (6.7%)	4 (8.9%)	
CC	2 (1.1%)		2 (4.5%)	0	0	0	
*MT1A* (rs11640851)		0.082					<0.001
CC	57 (32.0%)		24 (55.8%)	19 (42.2%)	10 (21.7%)	4 (9.1%)	
AC	77 (43.3%)		14 (32.6%)	13 (28.9%)	25 (54.3%)	25 (56.8%)	
AA	44 (24.7%)		5 (11.6%)	13 (28.9%)	11 (23.9%)	15 (34.1%)	
*MT1A* (rs8052394)		0.067					0.337
AA	74 (41.3%)		16 (36.4%)	17 (37.8%)	22 (47.8%)	19 (43.2%)	
AG	73 (40.8%)		16 (36.4%)	18 (40.0%)	18 (39.1%)	21 (47.7%)	
GG	32 (17.9%)		12 (27.3%)	10 (22.2%)	6 (13.0%)	4 (9.1%)	
*MT2A* (rs10636)		0.158					0.001
GG	101 (56.4%)		33 (76.7%)	29 (64.4%)	17 (37.0%)	22 (48.9%)	
GC	62 (34.6%)		10 (23.3%)	15 (33.3%)	21 (45.7%)	16 (35.6%)	
CC	16 (8.9%)		0	1 (2.2%)	8 (17.4%)	7 (15.6%)	
*MT2A* (rs28366003)		0.238					0.093
AA	150 (83.8%)		41 (95.3%)	37 (82.2%)	35 (76.1%)	37 (82.2%)	
AG	29 (16.2%)		2 (4.7%)	8 (17.8%)	11 (23.9%)	8 (17.8%)	
GG	0 (0%)		0	0	0	0	

* Chi-square and Fisher’s exact tests. The numbers of some genotypes do not add to 181 because of missing data.

**Table 4 ijerph-17-02909-t004:** Linear regression models of the association of neurological examination outcomes with TWICL exposure and gene polymorphisms among the lead workers. Only regression coefficients (standard error) that were significant (*p* < 0.05) are shown here.

	Hand VPT	Foot VPT	Hand CPT 5 Hz	Hand CPT 250 Hz	Hand CPT 2000 Hz	Foot CPT 5 Hz	Foot CPT 250 Hz	Foot CPT 2000 Hz
TWICL (no gene adjustment)	0.11 (0.05)	0.29 (0.10)	0.30 (0.14)			0.44 (0.19)	0.78 (0.31)	2.20 (0.68)
TWICL	0.13 (0.05)	0.30 (0.10)	0.34 (0.15)				0.75 (0.33)	2.18 (0.71)
*VDR*-Bsm rs1544410 (GA/GG)								
(AA/GG)								
TWICL	0.13 (0.05)	0.29 (0.10)				0.40 (0.19)	0.67 (0.32)	2.07 (0.70)
*VDR*-Apa rs7975232 (CC/AA)			10.45 (4.53)	17.23 (6.28)	31.83 (13.76)			
(AC/AA)								
TWICL	0.12 (0.05)	0.31 (0.10)					0.67 (0.33)	1.90 (0.71)
*VDR*-Taq rs731236 (TC/TT)	3.56 (1.64)							
(CC/TT)								
TWICL		0.21 (0.09)						1.80 (0.72)
*MT1A* rs11640851 (AA/CC)			12.92 (3.87)	12.60 (5.61)	36.58 (11.87)	12.20 (5.44)	25.26 (8.89)	63.89 (19.22)
(AC/CC)								
TWICL		0.17 (0.09)	0.33 (0.14)				0.68 (0.32)	2.00 (0.70)
*MT1A* rs8052394 (GG/AA)					−39.43 (11.91)			−40.52 (19.94)
(AG/AA)								
TWICL		0.29 (0.10)				0.42 (0.20)	0.73 (0.33)	2.05 (0.73)
*MT2A* rs10636 (GC/GG)			12.35 (2.98)	18.07 (4.15)	23.66 (9.10)			
(CC/GG)			15.05 (5.11)	23.36 (7.10)	60.02 (15.60)			
TWICL		0.26 (0.10)	0.41 (0.14)	0.48 (0.20)	1.09 (0.41)	0.51 (0.19)	0.85 (0.31)	2.45 (0.69)
*MT2A* rs28366003 (AG/AA)	3.28 (1.33)		−13.52 (3.82)	−19.10 (5.35)	−63.03 (10.94)	−19.57 (5.09)	−26.07 (8.51)	−51.42 (18.87)

All models were adjusted for age, sex, body height, smoking, alcohol, and working life span. Hand VPT and CPT were additionally adjusted for use of hand vibration tools.

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
