# Peer review of "Effects of Vitamin D Receptor, Metallothionein 1A, and 2A Gene Polymorphisms on Toxicity of the Peripheral Nervous System in Chronically Lead-Exposed Workers"

_ijerph, 2020, doi:10.3390/ijerph17082909_

Round 1

Reviewer 1 Report

This MS explores the relationship between polymorphisms of VDR, MT1A 305 and MT2A genes and the impact of lead toxicity on the peripheral nervous system in chronic lead–exposed workers. It is well organized and minor english language corrections are required.

Please reorganize the title to avoid the word “Lead” twice.

Line 46 – Please rephrase for a distinction between system and organs: ” The peripheral nervous system is one 46 of the target organs in lead intoxication”.

Please reorganize title of Table 1 and Table 3, since it is mentioned *The total numbers of education level, and genotypes are not 181 due to missing data.

Line 301 - Please rephrase: “The other limitation is that no multiple gene polymorphism analysis was performed, due to the relatively small sample size and the large number of possible gene combinations.”

Please consider the following paper:

Mani MS et al., Ecogenetics of lead toxicity and its influence on risk assessment. Hum Exp Toxicol. 2019 Sep;38(9):1031-1059.

Reviewer 2 Report

Review of the article: ijerph-727823

INTRODUCTION:

General comment: In my opinion the introduction should be significantly improved. Based on the title and content of the manuscript, separate paragraphs are not written on the same level. Meaning, the paragraph on MTs is written very detailed, also with some unnecessary information (location of genes on chromosome), while paragraph on the VDR is very scarce. The paragraphs should be more connected, argumentative and supported by the examples from the relevant literature data. For example, some information should be added on the connections between VDR, calcium and lead levels and consequently possible connection with peripheral neurotoxicity. It is not only that lead exposure effects calcium and vitamin D levels but also the other way around. It is known that calcium status effects lead absorption, lead binds on calcium proteins which are regulated by the vitamin D and VDR – therefore the genetic variations in genes related to vitamin D or calcium metabolism might influence internal (blood) levels of Pb and therefore its toxic effects. (reviewed in Broberg et al., 2015; DOI: 10.1016/B978-0-444-59453-2.00012-3).

Specific comments:

Line 49 – 50: no need for both measures of milk consumption; ~ 700ml per day is enough. Additionally, some further information on why milk might be beneficial against lead neurotoxicity (Calcium?) would be beneficial for the clarification and better connection with the next paragraph.

Line 54: not just bone concentration but also Pb in blood (plasma).

Line 55-56: This sentence should be combined with previous one as: Some authors suggest that variant vitamin D receptor (VDR) alleles (e.g. Bsm rs…, Apa rs…, Taq rs..)  modify lead concentrations in the blood and bone …..;  This paragraph should be supported by specific examples from existing literature (SNPs related with higher, lower Pb levels etc.).

Line 58: information on the identification of MT is not necessary.

Line 59: term metal(loid)s might be more suitable than heavy metals – the term “heavy metals” is misleading especially when considering also As.

Line 63: inducible by what, if meant by metal(loid)s it should be stated. Also, rephrase sentence, example: …..MT isoforms, consisting of sub-isoforms, coded by various functional genes.

Line 64-65: As stated above, information on chromosome location is not necessary.

Line 66: statement needs reference.

Line 67: exchange term genomic polymorphism with genetic or gene polymorphism.

Line 69-70: please rephrase sentence, example:  ….on the relationship between the polymorphisms of VDR, MT1A and MT2A and lead effects on the sensory nervous  system in humans. As both VDR and MT polymorphisms have been previously studied with the lead toxicity (neurodevelopment, hearing in children etc.)

METHODS: General comments: In the section on genetic analyses the order of methods should be revised. First it should be the part on DNA isolation and storage (which is now hidden inside MT1A, MT2A genotyping) - as it was used for genotyping of both VDR and MT SNP - then should follow the part on genotyping.

Specific comments:

Line 109-110: Were in the calculations (ICL, TWICL) included the measurements of Pb before 1990 although the reliable measurements began in 1991? What was the number or % of workers that had no previous records and were assigned value of 15µg/dl.

Line 137: please clarify the sentence - (are those 99% right handed from birth on, or they were thought to use the right hand).

Line 150: term occupation might be more suitable as job title.

Line 161 and 176 – 178: All studied genetic variations (VDRs & MTs) are single nucleotide polymorphisms just that VDR and MT SNPs were analysed using different method. Were dbSNP used to search only for MT SNPs? If yes it should be stated also how VDR SNPs were chosen; if not this paragraph should be moved before the description of the genotyping methods.  

Line 193: specify means (geometric or arithmetic). Following information are necessary: were the data on CPT, VPT, TWICL normally distributed (were log transformed data used in the models)?

Line 197: specify the grouping: …. according to quartiles of TWICL levels.

Line 198: Specify which statistical test was used to test for the difference of continuous variables (age, weight, height, working history) among TWICL groups (Table 1).

Line 199: specify the type of the regression analyses (multiple linear regression?)

RESULTS: General comments: The tables are in general well-constructed and clear (some specific corrections are stated below). However, the description of results is very scarce. The observed trends, differences should be described (e.g. For MT1A rs11640851 and MT2A rs…636: higher percentage of individuals with variant allele was observed in higher quartiles of TWICL levels;

Specific comments:

Line 210 – 211: Were 36 workers excluded? As in the table 1 characteristics are for all 181 participants. If it was meant the exclusion from regression analyses and table 2 that should be specified.

Line 214: Table 1 – Title, correct characters into characteristics; as there was a comparison between 4 groups it should be specified between which groups was the statistically significant difference.

Line 215 – 216: the sentence is ambiguous. Try to rephrase. (e.g. Statistically significant difference in blood Pb were observed for all tested neurological outcomes, except for hand VPT).  

Line 218: Table 2: Add number of participants for each test (as in table 1).; similarly, as for table 1 add among which groups was observed statistically significant difference.

Line 219: It would be useful to provide information on minor allele frequencies (MAF) of SNPs.

Line 222: Table 3: add % next to the total number of each genotype. Additionally, due to the small number of minor homozygotes in case of Bsm, Taq and MT2A rs…003, I would suggest grouping based on the presence or absence of minor allele (e.g. Bsm: GG vs (GA+AA) and repeat the Chi-square test as the results might differ.  Furthermore, there is a cconventional rule of thumb that if the expected numbers are below 5 alternative tests should be used (e.g. Fisher’s exact test).

Line 224: specify type of the regression analyses (multiple linear regression analyses).

Line 239: Table 4: were TWICL added in the model as groups of quartiles or as a continuous variable? Are results comparable between both options. Additionally, how were confounders that were used in the model chosen (why height and not BMI?).

225 – 236: Based on the aim of the study this paragraph should primarily discuss how the individual SNPs influenced the investigated association between TWICL and neurological outcomes and not mainly on associations of SNPs with outcomes. Furthermore,  I would recommend to additionally check the models where SNPs would be tested for the effect on TWICL levels (dependent variable) with the adjustment for confounders (age, gender….). This might add additional information for the discussion on possible effects of SNP on association between Pb and neurological outcomes.

DISCUSSION: General comment:  The discussion section should be revised to better connect the results of the manuscript with the supporting literature.

Specific comments

 Line 250: the abbreviation HbA1C should be explained as it is used for the first time.

Line 247 – 255: the comparison of regression coefficients between various studies might be tricky, as different statistical analyses (models) and data (log-transformed, not transformed) might be used between the studies.

Line 257 – 259: comment under table 3; statistical approach might not be appropriate; assumptions are made based on the very low number of AA individuals.

Line 264: add that the trend from the mentioned study was observed also in this manuscript although the difference was not significant.

Line 267: blood lead levels, internal lead dose – this is the same, use one of the terms

Line 279 - 281: specify the influence of MT1a rs…851 on CPT (protective or negative); Add information of higher % of A allele at lower TWICL levels to better implement hypothesis with Zn. Additionally, specify the effect of mentioned SNP on Zn levels (do AA genotypes have higher Zn levels)

Line 283: proposed models on testing influence of SNP on TWICL (mentioned in section results) - might be useful to support this statement;

Line 291: MT2A is rs10636 not 10635; intimately exchange with closely.

Line 308-309: this is difficult to say, as at the same time, the sensitivity for the lead induced neurological effects, increased with AG genotype.

Reviewer 3 Report

Introduction: line 44, please give the dates when lead was phased out…

Please frame the introduction by stating how bad lead is for human health and how  exposure comes from many sources:

Cite articles such as:

 Obeng-Gyasi, E., 2019. Sources of lead exposure in various countries. Reviews on environmental health, 34(1), pp.25-34.

Talk about lead exposure starting at pregnancy -

 Gulson, B. L., Jameson, C. W., Mahaffey, K. R., Mizon, K. J., Korsch, M. J., & Vimpani, G. (1997). Pregnancy increases mobilization of lead from maternal skeleton. Journal of Laboratory and Clinical Medicine, 130(1), 51-62.

Hu, H., Téllez-Rojo, M.M., Bellinger, D., Smith, D., Ettinger, A.S., Lamadrid-Figueroa, H., Schwartz, J., Schnaas, L., Mercado-García, A. and Hernández-Avila, M., 2006. Fetal lead exposure at each stage of pregnancy as a predictor of infant mental development. Environmental health perspectives, 114(11), pp.1730-1735.

- continuing during the life course:

Reuben, A., Caspi, A., Belsky, D.W., Broadbent, J., Harrington, H.,
Sugden, K., Houts, R.M., Ramrakha, S., Poulton, R. and Moffitt, T.E.,
2017. Association of childhood blood lead levels with cognitive function
and socioeconomic status at age 38 years and with IQ change and
socioeconomic mobility between childhood and adulthood. Jama, 317(12), pp.1244-1251.

And affecting several key organ systems within the human body.

Cardiovascular system

Obeng-Gyasi, E., 2019. Lead Exposure and Cardiovascular Disease among Young and Middle-Aged Adults. Medical Sciences, 7(11), p.103.

2)      Renal system

 Harari, Florencia, Gerd Sallsten, Anders Christensson, Marinka Petkovic, Bo Hedblad, Niklas Forsgard, Olle Melander et al. "Blood Lead Levels and Decreased Kidney Function in a Population-Based Cohort." American Journal of Kidney Diseases (2018).

 Lin, Ja-Liang, Dan-Tzu Lin-Tan, Kuang-Hung Hsu, and Chun-Chen Yu. "Environmental lead exposure and progression of chronic renal diseases in patients without diabetes." New England Journal of Medicine 348, no. 4 (2003): 277-286.

3)      Hepatic system

 Obeng-Gyasi, Emmanuel, Rodrigo X. Armijos, M. Margaret Weigel, Gabriel Filippelli, and M. Aaron Sayegh. "Hepatobiliary-Related Outcomes in US Adults Exposed to Lead." Environments 5, no. 4 (2018): 46.

Can, S., C. BaÄŸci, M. Ozaslan, A. I. Bozkurt, B. Cengiz, E. A. Cakmak, R. KocabaÅŸ, E. KaradaÄŸ, and M. TarakçioÄŸlu. "Occupational lead exposure effect on liver functions and biochemical parameters." Acta Physiologica Hungarica 95, no. 4 (2008): 395-403.

Only then can you start talking about its effects on the nervous system. This will make for a much more interesting read.

Methods:

What was the statistical power of the study? Do you have enough power to detect the effects you are measuring?

Results:

Please reformat all tables 1,2,3. Too much clutter and hard to read.

Discussion:

Very well done!

Round 2

Reviewer 2 Report

General comment: Manuscript has improved substantially and, in my opinion, provides interesting results for other scientists from similar field. However, for the acceptance in journal minor revision is still needed. Although, English improved there are still some sentences that need language correction. Within my specific comments below, I gave some suggestions how to rephrase such sentences, although I am not a native English speaker, so I would still suggest article to be corrected by a professional. Additionally, some issues in the discussion must be clarified.

INTRODUCTION:

Line 46:  I would suggest changing lead exposure causes into:  lead exposure is associated with several… There are still inconsistencies between various epidemiological studies therefore to use causality might be too strong.

Line 62: Please write out ICL abbreviation as it is used for the first time (explained latter in line 101)

Line 68: Maybe rephrase: MT1 and MT2A are isoforms….. (no need to say MT isoforms, as that is clear from paragraph above)

Line 75: The sentence “As both VDR and MT polymorphisms have been previously studied with the lead toxicity (neurodevelopment, hearing in children etc)” can be deleted; in my previous review this sentence was just the example for my correction of the sentence before (this is the first study on the relationship between the polymorphisms of the VDR, MT1A, and MT2A and lead effects…). Therefore, there is no need for this sentence in line 75, without it will be just fine. Sorry for the misunderstanding.

MATERIAL and METHODS:

Line 84: rephrase personal denied into either personally denied, or personal denial.

Line 89: information on total number of workers is already given in Line 84 (I would delete in line 84 and leave it in 89). 

Line 112: rephrase, there were 45% of the workers who entered…

Line 115: rephrase, into: this value was based….or  This was based on….

Line 167: To have consistence with line 182, maybe is better to rephrase title into: VDR genotyping

Line 187 and 194: This is confusing as PCR systems are written differently? Are there two systems? If you meant software, then sentence 187 should follow sentence 194. Or choose one description of the system.

RESULTS:

Line 215 - 216: the sentence: Due to the time required…., 36 workers did not undergo….is not necessary here as it is explained in the materials and methods.

Line 222-223: Sentence is confusing please rephrase: Similarly, body weight and working history were significantly the highest in the group of 75-100 percentile…

Line 230-231: This sentence is confusing and not understandable. If I understand correctly the purpose of the sentence please rephrase as: Significantly higher thresholds for foot VPT and both foot and hand CPT at all frequencies, were observed for individuals in the highest TWICL quartile (75-100 percentile).

233: It table 2: * ANOVA and post-hoc Scheffe tets

Line 235: Please rephrase as follow: The genotype distribution of three VDR, two MT1A and two MT2A SNPs, within each TWICL quartile, are shown in Table 3.

Line 236: There is no need to specify AA, CC, CC etc.. as with stating minor allele frequency is clear which allele it is. In case you still want to specify then VDR-Bsm AA 11.7% is incorrect because AA is genotype while 11.7% is the frequency of A allele occurring in AA and GA genotype; you can say A allele of VDR-Bsm.  I suggest to rephrase as: The minor allele frequencies for these SNPs are 12% for VDR-Bsm, 32% for VDR-Apa etc…

Line 238: rephrase: No SNPs significantly violated Hardy-Weinberg Equilibrium or All SNPs were in accordance with HWE.

Line 259: Table 4:  Rephrase: Only regression coefficients (standard error) which were significant (< 0.05) are shown here.

Did you check what is the actual effect of SNPs on levels of TWICL with the adjustment for confounders  (multiple linear regression models)?

DISCUSSION:

Line 270: rephrase: Very few studies discussed the influence of VDR-Apa (rs…) SNP.

Line 271: higher TWICL was noted for CC genotype

Line 272: Such trend was observed also….

Line 273-274: I suggest rephrasing: Moreover, all three frequencies of hand CPT were significantly higher for CC versus AA genotype of Apa.

Line 287: Rephrase: The G allele of MT2 rs…., compared to A allele

Line 279 and 285: How do you explain that G allele of MT1A is protective due to its anti-oxidative effect if at the same time is supposed to DECREASE the SOD anti-oxidativeactivity?

Line 287-293: This paragraph needs to be clarified. It seems that G allele of MT2A rs..003 is at the same time protective for neurotoxicity (based on negative coefficients, for almost all outcomes) and increases susceptibility for toxicity of lead due to SLIGHTLY increases in TWICL coefficients compared to TWICL coefficients without SNP included. I would include following: “Based on the much higher coefficient of SNP than of TWICL , this would suggest that G genotype has more protective effect towards neurotoxicity and as you suggested AA represent susceptibility for neurotoxicity. As such I would move sentence in line 288 at the end of this paragraph, as the literature cited is related to first sentence.

Line 292: here the suggestion on protective effect of Zn needs clarification, based on what is suggested protective effect - based on low Zn levels at GG genotype?

Line 297: specify influence: negative influence

Line 304: significant results

CONCLUSIONS

Line 309: in chronically lead-

Line 311 also 288: it is AA genotype

Author Response

Thanks for the suggestion. Please see the attachment

Reviewer 3 Report

Significant improvement, the article is ready for publication.

Author Response

Thanks for the suggestion. We have rephrased some sentences that need language correction.